# Vine Physiology, Yield Parameters and Berry Composition of Sangiovese Grape under Two Different Canopy Shapes and Irrigation Regimes

Gabriele Valentini, Chiara Pastore *, Gianluca Allegro, Riccardo Mazzoleni, Fabio Chinnici and Ilaria Filippetti

Department of Agricultural and Food Sciences, University of Bologna, Viale Fanin 44, 40127 Bologna, Italy
* Correspondence: chiara.pastore@unibo.it

**Abstract:** *Vitis vinifera* L. adapts well to a scarce availability of water in the soil. However, in recent decades, the combination of thermal stress with prolonged water scarcity could have dramatic consequences on the vine's physiological status. In this paper, we evaluated the effects of two canopy shapes and two irrigation regimes at veraison on vine physiology, yield parameters and grape composition through biochemical and molecular approaches. The water shortage strongly influenced the physiology of Sangiovese only when the stress was moderate to severe. Neither the water stress limited to veraison nor the canopy shape were able to influence the yield parameters and sugar content, and a strong induction of the expression of the genes involved in the biosynthesis of anthocyanins was recorded only in conditions of moderate-to-severe stress. This phenomenon led to an increase in the anthocyanin content in berry skins until the end of veraison. Conversely, no significant effects occurred in terms of biochemical and molecular performance after re-watering and at harvest. Though the shape of the canopy could play a role only under elevated temperature and prolonged drought, severe water stress can affect the vine physiology and berry ripening during the veraison stage.

**Keywords:** climate change; grapevine; drought; training system; anthocyanin; gene expression





## 1. Introduction

Over the past few decades, climatic conditions have dramatically changed worldwide, giving rise to severe transformations in the most relevant grape-growing areas [1,2]. Moreover, extended summer droughts and rising temperatures are increasingly forecasted [3,4]. The combination of excessive photosynthetically active radiation (PAR), UV radiation and temperature might be exacerbated by scarce water availability, and, in our environment, this condition becomes critical especially at the onset of ripening [5,6]. Sangiovese, the most widespread grape variety in Italy [7], is considered a low-coloured grape when compared to the average anthocyanin concentrations observed in several grapevine cultivars and has a composition mainly represented by glycosylated forms [8,9]. Veraison is a crucial stage for bearing out proper grape ripening, and it is well-established that at this stage, anthocyanin accumulation is promoted by sunlight exposure [10]. In several Sangiovese viticultural areas, grapes are often exposed to sunlight radiation during veraison due to the aging of basal leaves [11]. Consequently, this might lead to a temperature excess around the grape bunch microclimate, which may be responsible for the detriment of anthocyanin accumulation [12–14] due to the promotion of oxidative enzymes, such as peroxidases (PODs) [15], and the minor expression efficiency of genes involved in biosynthesis [16].

Moreover, the negative effects of high temperatures might be further enlarged and paralleled with a drastic decrease in soil water content [5,17,18]. *Vitis vinifera* L. is among the species reputed to be versatile for its high adaptability to low water availability [19]. Nevertheless, the combination of thermal stress and lengthened water shortage might

have drastic consequences on the vine's physiological status due to the unbalanced plant's evaporative demand and soil water content [20,21]. Schultz [22] identified different hydraulic features of each grape genotype that appear to control their stomatal behavior under water stress. Accordingly, Tombesi et al. [23] proposed that this should be represented as a continuum of genotype behaviors with different levels of anisohydry. In this regard, Sangiovese is considered to have a near-anisohydric behavior that tends to regulate the water potential by reducing the transpiration and optimizing water use efficiency, normally with positive implications in terms of grape production and quality [24,25].

Several vineyard management techniques may represent interesting tools to modulate a condition of multiple types of stress (hydric, thermal and radiative) that often turn up in our environment during the onset of the Sangiovese's ripening. Among these, the vine training system and canopy architecture may strongly affect both the bunch's sun exposure [26–28] and the canopy's light interception, with consequences on berry ripening and physiological vine performance. Moreover, the quantity of intercepted light as conditioned by the canopy architecture is one of the most important determinants of vine water use in the grapevine [29]. Therefore, the distinction between opened- rather than closed-canopy architecture plays a crucial role in the vine's physiological status and in light interception [26]. It has been reported that closed-canopy-trained vines intercept a greater fraction of light during cooler hours of the morning, whereas opened-canopy ones have the greatest light incidence from the early afternoon, during the highest temperature of the day [27,30]. The effect of the vine's architecture on canopy radiation distribution and grape production has been largely studied, as have the effects of the canopy shape on leaf gas exchange and water use efficiency (WUE) [27,29,31–37]. However, in agreement with Naulleau et al. [28], due to scant comparison among the water use efficiency of different training systems, it is difficult to state which one is better adapted to drought.

Based on these considerations, this study aimed to evaluate how potted vines of Sangiovese trained with opened and closed canopies might respond to the implementation of controlled water stress around veraison while also considering their different light interception due to canopy architecture. To accomplish this, the vine's physiological, biochemical and biomolecular analyses were assessed throughout the 2018 and 2019 growing seasons.

## 2. Materials and Methods

### 2.1. Plant Material and Experimental Design

The trial was carried out during the 2018 and 2019 seasons on 16 uniform potted *Vitis vinifera* L. cv. Sangiovese (clone 12T) grapevines grafted onto SO4 rootstock at the Department of Agricultural and Food Sciences (DISTAL) of the University of Bologna (Italy). The vines were planted in 2010 in 30 L pots filled with a soil mixture (39% sand, 39% silt and 22% clay) with an organic matter content of 1.8% and a pH of 7.8. Field capacity and wilting point were calculated after Saxton and Willey (2005) and set at 0.29 $cm^3/cm^3$ and 0.14 $cm^3/cm^3$, respectively. The vines were arranged on two different training systems and canopy shapes on a single N-S oriented row: a Guyot vertical shoot-positioned (C) and V-shaped open-canopy grapevine (V) with a vine spacing of 1 m along the row. Each vine was pruned to two fruiting canes of 10 buds. The fruiting canes were 0.8 m from the ground. In C, all the shoots were constrained in a trellising system consisting of two pairs of catch wires, and in V, the canopy takes a V shape thanks to a frame with an overall aperture angle of 80°. Trimming was not performed during the season, and vines were provided with mineral nutrition based on the yearly consumption of the vine. The 16 vines were assigned to well-watered (WW) and water-stressed treatments (WS). All the vines were kept well-watered until the day-of-year (DOY) 211 (30 July) in 2018 and DOY 217 (5 August) in 2019, which represent the onset of veraison, providing the amount of water lost through transpiration—approximately 4 Lday$^{-1}$ of water (WW)—automatically distributed via a drip irrigation system. Starting from the above dates, the vines afferent to the stressed treatments (WS) were irrigated at 50% of WW up to DOY 222 (10 August) in 2018 and until DOY 232 (20 August) in 2019, which represent the end of the veraison stage. Furthermore,

the pots were entirely covered with aluminum foil before the trial started in order to avoid overheating. Throughout the growing season, air temperature and rainfall were monitored by an automatic meteorological station close to the vineyard.

### 2.2. Physiological Measurements, Vegetative Evaluation and Yield Data

Leaf assimilation rate (An) and stomatal conductance (gs) were measured using a portable gas exchange Li-Cor 6400 system (Li-Cor Inc., Lincoln, NE, USA) on three well-exposed mature principal leaves inserted between nodes 8 and 12. Intrinsic water use efficiency ($WUE_i$) was then calculated as An/gs ratio. Single-leaf gas exchange measurements were taken around midday on DOY 215 and 222 in 2018, and in 2019, on DOY 219, 225 and 231 using a broadleaf chamber under a saturated light (1500 $\mu$mol m$^{-2}$s$^{-1}$) using an external lamp. At harvest (DOY 257 in 2018 and DOY 256 in 2019), yield attributes were assayed, e.g., crop weight, cluster number and pruning weight were measured for each vine in both years during the winter. To account for precise vine water use, individual daily gravimetric vine water loss (E) was continuously measured throughout the growing season using a platform scale mod. LAUMAS (ABC Bilance, Campogalliano, MO, Italy) was placed underneath each pot. Pot masses were recorded every ten minutes using a CR1000 datalogger (Campbell Scientific, Inc., Logan, UT, USA). Each pot surface was covered with a plastic film to avoid interference from rainwater and to minimize losses due to soil evaporation. Each pot was refilled with water distributed automatically using a drip irrigation system during the hours of minimal transpiration. Midday stem water potential ($\Psi_{stem}$) was assayed on 16 vines using a Scholander-type pressure chamber (mod. 3005, Soil Moisture Equipment Corp., Santa Barbara, CA, USA) at 1 pm on a mature leaf per vine. The measurements were performed in August 2018 on DOY 213, 215, 222 and 242, and in 2019, on DOY 219, 225, 226, 231 and 247.

### 2.3. Must Analyses and Anthocyanin, Flavonols and Stilbenes Separation via HPLC

In the same days of physiological measurements and at harvest, in both years, 10 berries were collected from each vine for the analysis of °Brix, pH and titratable acidity. At the same time, 20 berries per vine were collected and total anthocyanins were extracted from the skins of the berries by soaking the peeled skins in 100 mL methanol for 24 h, then storing the extracts at $-20$ °C [8]. Total and extractable anthocyanins were separated by high-performance liquid chromatography (HPLC) using a Waters 1525 instrument equipped with a diode array detector (DAD) and a reversed-phase column (RP18 250 × 4.6 mm, 5 $\mu$M) with a pre-column (Phenomenex, Castel Maggiore, BO, Italy). Anthocyanins were quantified at 520 nm using an external calibration curve with malvidin-3-glucoside chloride as the standard (Sigma-Aldrich, St. Louis, MO, USA). In 2019, flavonol and stilbene analysis was conducted on the same anthocyanins extracts [8], which had undergone HPLC separation according to Castro Marin and Chinnici [38]. Samples were directly injected without pre-treatment except filtration. The HPLC instrument was equipped with a quaternary gradient pump Jasco PU-2089, an autosampler Jasco AS-2057 Plus Intelligent Sampler and two detectors: a Jasco UV/Vis MD-910 PDA detector and a Jasco FP-2020 Plus Fluorescence detector (Jasco, Tokyo, Japan). The column was a C18 Poroshell 120 (Agilent Technologies, Santa Clara, CA, USA), 2.7$\mu$m, (4.6 mm × 150 mm), operating at 35 °C with a flow of 0.8 mL/min. Elution solvents were 2% acetic acid in HPLC-grade water (Eluent A) and 2% acetic acid (Sigma-Aldrich, St. Louis, MO, USA) in HPLC-grade acetonitrile (Sigma-Aldrich) (Eluent B). Gradient elution was as follows: from 98% to 95% A in 10 min, 95% to 90% A in 7 min, 90 to 82% A in 6 min, 82% to 80% A in 3 min, 80% to 70% A in 3 min, 70% to 50% A in 3 min, 50% to 0% A in 4 min and finally to 98% A in 1 min. Quantification was performed by means of calibration curves previously obtained by duplicate injections of pure standards solutions at known concentrations. Standard compounds (Extrasynthese, Genay, France) were: myricetin-3-glucuronide, myricetin-3-glucoside, myricetin-3-rhamnoside, quercetin-3-glucuronide, quercetin-3-glucoside, laricitrin-3-glucoside, kaempherol-3-glucoside, syringetin-3-glucoside,

myricetin, quercetin, *t*-resveratrol and *t*-resveratrol-3-*O*-glucoside. All the analyses were carried out in triplicate.

### *2.4. RNA Extraction and Gene Expression Analysis*

With the exception of the harvest, 20 berries were sampled from each vine on each sampling date, and RNA extraction was performed on berry skin using the Spectrum Plant Total RNA kit (Sigma-Aldrich). RNA quality and quantity were determined using a Nanodrop-1000 spectrophotometer (Thermo Scientific, Wilmington, DE, USA). One microgram of extracted RNA was treated with two units of DNase I (Promega, Madison, WI, USA) and then reverse-transcribed using Improm-II Reverse Transcriptase (Promega, USA) according to the manufacturer's instructions. Real-time quantitative PCR analysis was performed with a dilution of cDNA (1:20), to which a master mix containing SYBR Green (Applied Biosystems, Foster City, CA, USA) and the primers of the genes of interest were added. The PCR reaction was conducted on an ABI PRISM Step One Plus system (Applied Biosystems, Foster City, CA, USA), as reported in Pastore et al. [39]. Non-specific PCR products were identified by dissociation curves. Each reaction was performed in three technical replicates, using actin and ubiquitin as housekeeping genes. The expression of all the genes involved in the first and late steps of anthocyanin biosynthesis was assessed, and their primers were retrieved from the literature: PAL1 (phenylalanine ammonia-lyase) in Belhadj et al. [40]; DFR (dihydroflavonol reductase), LDOX (leucoanthocyanidin dioxygenase) and UFGT (UDP-glucose: flavonoid 3-O-glucosyl transferase) from Goto-Yamamoto et al. [41]; and MYBA1 from Jeong et al. [42]. Amplification efficiency was calculated with the LinRegPCR software and used in the calculation of the MNE (Mean Normalized Expression), as reported in Simon [43]. The mean normalized expression (MNE) value was calculated for each sample referring to the actin and ubiquitin expressions according to the Simon equation.

### *2.5. Statistical Analysis*

All the data were analyzed by the mixed procedure available in Sas v9.0 (SAS Institute, Inc., Cary, NC, USA). For gene expression data, one-way analysis of variance (ANOVA) was conducted. Treatment comparisons were analyzed using the Tukey test for pairwise comparison with mean separation by $p < 0.05$.

## 3. Results

### *3.1. Weather Conditions Recorded during the Years of Study and during the Water Stress Treatment*

The average maximum, minimum and mean temperatures as well as rainfalls recorded during the growing season (April–October) in 2018 and 2019 are shown in Table 1.

**Table 1.** Mean, minimum and maximum temperatures and rainfalls recorded throughout the 2018 and 2019 growing season (April–October).

| | 2018 | | | | 2019 | | | |
|---|---|---|---|---|---|---|---|---|
| | T Mean (°C) | T Min (°C) | T Max (°C) | Rainfall (mm) | T Mean (°C) | T Min (°C) | T Max (°C) | Rainfall (mm) |
| April | 16.7 | 12.2 | 21.4 | 18.6 | 14.3 | 8.8 | 19.7 | 53.6 |
| May | 19.3 | 15.3 | 24.2 | 59.4 | 15.3 | 10.9 | 19.9 | 185.5 |
| June | 23.2 | 18.3 | 28.7 | 97 | 25.2 | 18.4 | 31.4 | 24.4 |
| July | 26.0 | 20. 8 | 31.6 | 49.2 | 25.8 | 19.1 | 32.1 | 30.2 |
| August | 26.1 | 20.8 | 31.7 | 28.2 | 25.6 | 19.3 | 32.3 | 30.5 |
| September | 22.1 | 16.8 | 26.9 | 32.2 | 20.4 | 15.0 | 26.1 | 48.9 |
| October | 16.7 | 12. 8 | 21.1 | 61.8 | 16.6 | 12.1 | 21.1 | 45 |

During veraison (the period in which the water stress was applied), the climatic conditions were different between the two tested years (Figure 1). The average air temperature was around 3 °C higher in 2018 than in 2019, and relative air humidity was higher in 2019.

The climatic conditions implied that in 2018, veraison lasted 10 days and in 2019, it lasted 15 days.

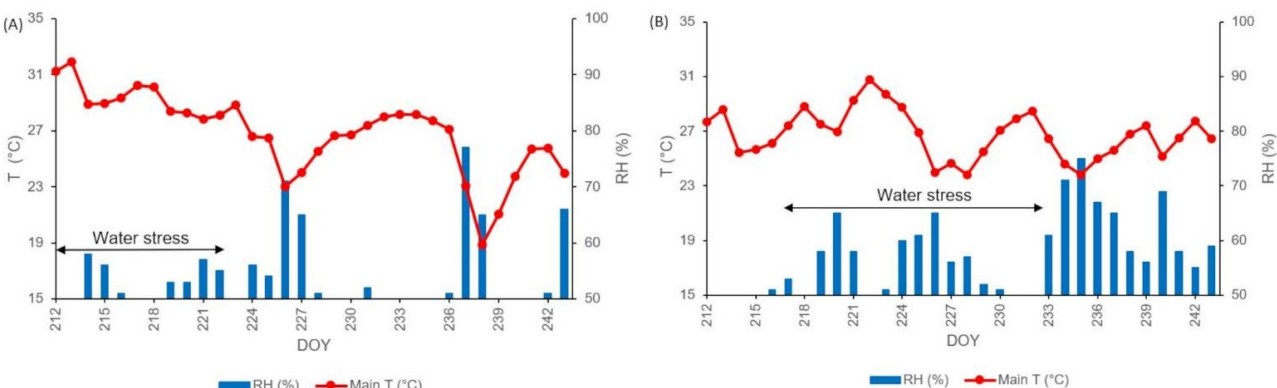

**Figure 1.** Daily air temperature and relative humidity during the month of August in the 2018 (**A**) and 2019 (**B**) seasons. The duration of the water stress period is indicated by black double arrows.

### 3.2. Physiological Analyses and Yield Data at Harvest

The data in Figure 2 represent the transpiration of the vines in the water restriction period between the end of July and the first half of August. It should be noted that the 50% reduction in water supply for the water-stressed (WS) vines resulted in a consequent lowering of the transpiration rate compared to the well-watered (WW) ones. Though there was a clear and significant separation between WW and WS transpiration curves in 2018 (Figure 2A), no differences were detected in terms of water loss between rent training systems (data not shown). In 2019, there was a clear detachment (Figure 2B) between the transpiration rates of WW and WS just one week after the induction of water stress, whereas the difference in the transpiration curves of C and V were not statistically significant (data not shown). The analysis of the stem water potential ($\Psi_{stem}$) highlighted both the different water status of the irrigated vines compared to the stressed ones and the duration of the water restriction over the season (Table 2). C and V did not show significant differences in terms of water status in the two-year trial. However, stem water potential of WS vines progressively decreased along with the continuous reduction in water supply. In 2018, the WS theses reached a more negative stem potential the day before re-watering and twenty days after re-watering on 30 August (DOY 242), recovering to the same level of WW vines. In 2019, the value of severe water stress for the vine was instead reached nine days after water reduction. As shown in Figure 1, the wetter climatic conditions resulted in both lower evaporative demand and better water status of the vines until 14 August (DOY 226). As for the year 2018, in the 2019 season, at the recovery day (9 September, DOY 252), there was no difference between the theses in terms of $\Psi_{stem}$. Though the assimilation rate (An) did not differ between training systems, photosynthetic limitation was found in WS treatment during veraison in 2018. In 2019, only under intense water stress reached on 19 August (DOY 231) did WS show a significant reduction in carbon fixation by the leaves (Table 3). In detail, the interaction between the two main factors (training system × water shortage) showed that in 2018, the photosynthetic assimilation for C was greater than in the other theses. The evaluation of stomatal conductance (gs) during the period of induced water stress showed a substantial effect of water scarcity with respect to the training system (Table 4). Since the passive mechanisms of resistance to abiotic stress, such as stomatal control, are particularly effective in limiting the embolism of xylem vessels, a strong reduction in gs was highlighted as the water potential approached values considered critical (Tables 2 and 4). In 2018, the WS treatment showed about a 70% reduction in stomatal activity compared to WW. This reduction was significantly less marked, though significant, in 2019. As for assimilation and stomatal conductance, the interaction between vine shape and water restriction has highlighted some significant

findings. In 2018, C separated very well from A and from the two theses subjected to water stress (Tables 3 and 4). With reference to the water-use efficiency (WUEi, Table 5), it can be noted that in 2018, the An/gs ratio was higher in the non-irrigated theses that optimized $CO_2$ absorption compared to water loss. This only happened in 2019 after two weeks of irrigation restrictions and just before re-watering. Starting from the same number of clusters, no differences were found at harvest in terms of yield per vine, berry size and pruning weight in either 2018 or 2019 (Table 6).

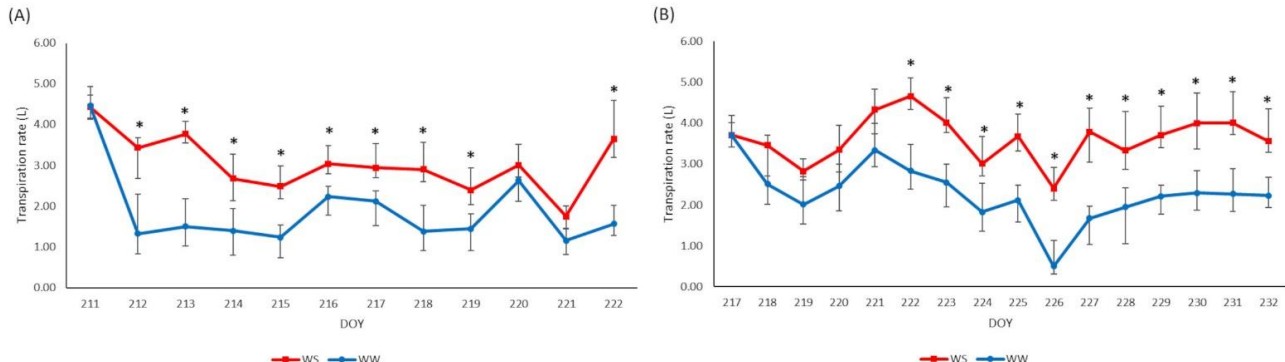

**Figure 2.** Seasonal pattern of daily lysimeter transpiration in water-stress (WS) and well-watered (WW) vines during the stress period in the 2018 (**A**) and 2019 (**B**) seasons. Data are average of 8 vines ± SE. Within each day, an asterisk indicates a significant difference between the two water regimes as calculated by the Tukey test ($p \leq 0.05$).

**Table 2.** Stem water potential ($\Psi_{stem}$, bar) of Sangiovese vines subjected to both a different training system and water restriction during the veraison stage. The re-watering date was DOY 242 and 247 for the 2018 and 2019 season, respectively. Values represent means of 4 replicates: C WW = well-watered vertical shoot-positioned canopy, V WW = well-watered V-shaped open canopy, C WS = vertical shoot-positioned canopy with water restriction, V WS = V-shaped open canopy with water restriction. Significant differences are indicated: *, $p < 0.05$; ns, not significant. Mean values were separated within each column with different letters using the Tukey test.

| | 2018 | | | | 2019 | | | |
|---|---|---|---|---|---|---|---|---|
| **DOY** | **213** | **215** | **222** | **242** | **219** | **225** | **231** | **247** |
| C WW | −8.5 | −7.5 | −6.4 | −7.2 | −5.7 | −5.7 | −7.6 | −6.1 |
| V WW | −8.3 | −7.6 | −7.1 | −7.7 | −6.5 | −6.1 | −6.7 | −6.5 |
| C WS | −11.5 | −13.0 | −13.5 | −7.4 | −6.8 | −6.6 | −8.8 | −6.4 |
| V WS | −11.3 | −12.5 | −16.0 | −6.5 | −7.8 | −7.7 | −9.7 | −6.1 |
| *Training system effect* | ns | ns | ns | ns | ns | ns | ns | ns |
| *Water stress effect* | * | * | * | ns | ns | ns | * | ns |
| *Significance of training system × water stress interaction* | ns | ns | ns | ns | ns | ns | ns | ns |

**Table 3.** The leaf assimilation rate (An, $\mu$mol m$^{-2}$s$^{-1}$) during the 2018 and 2019 season. Values represent means of 4 replicates: C WW= well-watered vertical shoot-positioned canopy, V WW = well-watered V-shaped open canopy, C WS = vertical shoot-positioned canopy with water restriction, V WS = V-shaped open canopy with water restriction. Significant differences are indicated: *, $p < 0.05$; ns, not significant. Mean values were separated within each column with different letters using the Tukey test.

|  | **2018** | | **2019** | | |
| :---: | :---: | :---: | :---: | :---: | :---: |
| **DOY** | **215** | **222** | **219** | **225** | **231** |
| C | 17.0 a | 13.0 | 17.6 | 11.5 | 14.6 |
| V | 11.9 b | 12.9 | 16.5 | 11.6 | 14.0 |
| C WS | 6.4 c | 6.4 | 14.4 | 11.4 | 12.6 |
| V WS | 8.0 bc | 7.5 | 15.8 | 10.6 | 13.1 |
| *Training system effect* | ns | ns | ns | ns | ns |
| *Water stress effect* | * | * | ns | ns | * |
| *Significance of training system × water stress interaction* | * | ns | ns | ns | ns |

**Table 4.** Leaf stomatal conductance (gs, mol m$^{-2}$s$^{-1}$) during the 2018 and 2019 seasons. Values represent means of 4 replicates: C WW= well-watered vertical shoot-positioned canopy, V WW = well-watered V-shaped open canopy, C WS = vertical shoot-positioned canopy with water restriction, V WS = V-shaped open canopy with water restriction. Significant differences are indicated: *, $p < 0.05$; ns, not significant. Mean values were separated within each column with different letters using the Tukey test.

|  | **2018** | | **2019** | | |
| :---: | :---: | :---: | :---: | :---: | :---: |
| **DOY** | **215** | **222** | **219** | **225** | **231** |
| C WW | 0.23 a | 0.22 | 0.20 | 0.17 | 0.16 |
| V WW | 0.17 b | 0.25 | 0.18 | 0.14 | 0.15 |
| C WS | 0.07 c | 0.07 | 0.16 | 0.12 | 0.12 |
| V WS | 0.07 c | 0.08 | 0.17 | 0.12 | 0.11 |
| *Training system effect* | ns | ns | ns | ns | ns |
| *Water stress effect* | * | * | * | * | * |
| *Significance of training system × water stress interaction* | * | ns | ns | ns | ns |

**Table 5.** WUEi (An/gs) during the 2018 and 2019 seasons. Values represent means of 4 replicates. C WW = well-watered vertical shoot-positioned canopy, V WW = well-watered V-shaped open canopy, C WS = vertical shoot-positioned canopy with water restriction, V WS = V-shaped open canopy with water restriction. Significant differences are indicated: *, $p < 0.05$; ns, not significant. Mean values were separated within each column with different letters using the Tukey test.

|  | **2018** | | **2019** | | |
| :---: | :---: | :---: | :---: | :---: | :---: |
| **DOY** | **215** | **222** | **219** | **225** | **231** |
| C WW | 70 | 59 | 88 | 67 | 91 |
| V WW | 74 | 52 | 92 | 84 | 93 |
| C WS | 91 | 91 | 90 | 95 | 105 |
| V WS | 114 | 94 | 93 | 90 | 119 |
| *Training system effect* | ns | ns | ns | ns | ns |
| *Water stress effect* | * | * | ns | ns | * |
| *Significance of training system × water stress interaction* | ns | ns | ns | ns | ns |

**Table 6.** Yield data and pruning weight at harvest. Data are collected in 2018 and 2019 seasons. Values represent means of 4 replicates: C WW= well-watered vertical shoot-positioned canopy, V WW = well-watered V-shaped open canopy, C WS = vertical shoot-positioned canopy with water restriction, V WS = V-shaped open canopy with water restriction. Significant differences are indicated: ns, not significant. Mean values were separated within each column with different letters using the Tukey test.

| | 2018 | | | | 2019 | | | |
|---|---|---|---|---|---|---|---|---|
| | Bunches (n°) | Yield (kg) | Berry Weight (g) | Pruning Weight (kg) | Bunches (n°) | Yield (kg) | Berry Weight (g) | Pruning Weight (kg) |
| C WW | 10 | 0.79 | 1.84 | 355 | 15 | 0.96 | 1.23 | 308 |
| V WW | 11 | 0.93 | 1.87 | 285 | 15 | 0.91 | 1.08 | 268 |
| C WS | 10 | 0.56 | 1.58 | 410 | 15 | 0.83 | 1.09 | 285 |
| V WS | 11 | 0.63 | 1.55 | 320 | 15 | 0.81 | 1.05 | 298 |
| *Training system effect* | ns | ns | ns | ns | ns | ns | ns | ns |
| *Water stress effect* | ns | ns | ns | ns | ns | ns | ns | ns |
| *Significance of training system × water stress interaction* | ns | ns | ns | ns | ns | ns | ns | ns |

### 3.3. Berry Composition and Anthocyanin Accumulation during Ripening and Phenol Profiles at Harvest

No differences in berry weight were recorded during ripening among treatments (data not shown) or in the trends of the total soluble solids concentration (TSS) either in 2018 and 2019 (Figure 3). Figure 3 shows that in both 2018 and 2019, during the stress period, the WS theses underwent an increase in the concentration of soluble solids compared to the irrigated ones, though this trend was not significant.

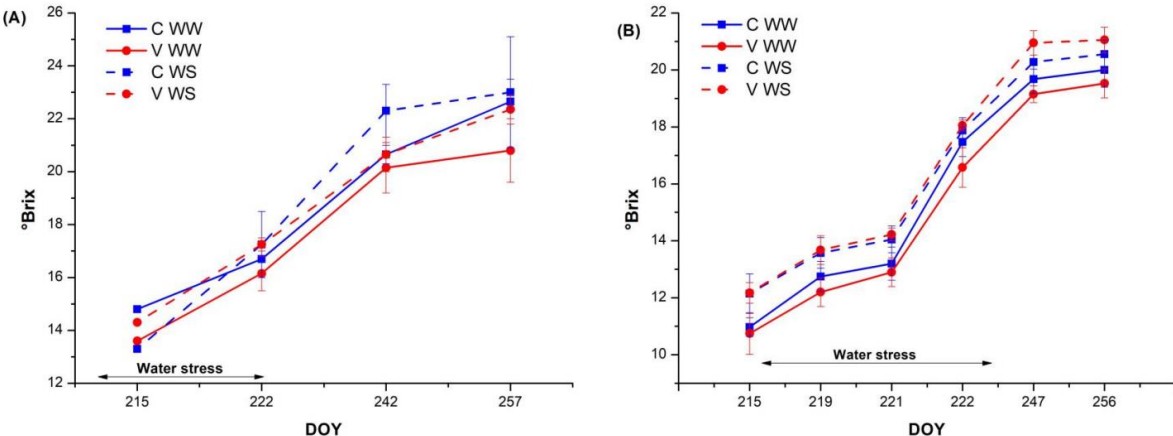

**Figure 3.** Evolution of total soluble solid content (°Brix) in Sangiovese berries sampled in closed (C) and V (V) canopy-shaped vines under normal water supply (WW) or water stress at veraison (WS) in 2018 (**A**) and 2019 (**B**) seasons. Error bars indicate the mean SE (n = 4). Means followed by different letters differ significantly, as calculated by Tukey test ($p \leq 0.05$). The duration of the water stress period is indicated by black double arrows.

Furthermore, at harvest and in both years, the concentration of soluble solids, titratable acidity and pH were not affected by the treatments (Table 7).

In each year, water stress induced a larger increase in the biosynthesis of anthocyanins compared to WW, regardless of canopy shape (Figure 4). However, this increase became statistically significant immediately after the application of the stress only in 2018 (Figure 4A), whereas in 2019, we found differences among treatments only at the end of stress period (Figure 4B). At recovery and until harvest, anthocyanin increase was confirmed only in the V WS treatment, as a reduction or slowdown in their accumulation was detected in the

berries of the C WS vines in both years. The behavior of the berries from non-stressed plants in terms of anthocyanin accumulation was also remarkably interesting. In the case of the V-canopy, after a delay in accumulation, which, in both years, coincided with the veraison phase, the anthocyanins showed an increase in concentration, which at the end of the season led to values close to those of the other theses and of the V WS berries. Otherwise, in C WW, after an initial acceleration, a decrease in anthocyanin accumulation occurred, and at harvest, C WW berries presented tendentially lower values of anthocyanin concentration in comparison to all the other treatments.

**Table 7.** Influence of treatments on the standard parameters of the must and on the polyphenolic content of Sangiovese vines in the 2018 and 2019 seasons. Values represent means of 4 replicates: C WW = well-watered vertical shoot-positioned canopy, V WW = well-watered V-shaped open canopy, C WS = vertical shoot positioned canopy with water restriction, V WS = V-shaped open canopy with water restriction. Significant differences are indicated: *, $p < 0.05$; ns, not significant. Mean values were separated within each column with different letters using the Tukey test.

| | 2018 | | | | 2019 | | | | | |
|---|---|---|---|---|---|---|---|---|---|---|
| | Soluble Solids (°Brix) | Total Acidity (g/L) | pH | Total Antho-cyanins (mg/kg) | Soluble Solids (°Brix) | Total Acidity (g/L) | pH | Total Antho-cyanins (mg/kg) | Total Flavonols (mg/kg) | Total Stilbenes (mg/kg) |
| C WW | 22.7 | 4.89 | 3.7 | 708.8 | 20.0 | 6.47 | 3.5 | 728.5 | 240.1 | 4.60 |
| V WW | 20.8 | 5.39 | 3.6 | 764.2 | 19.5 | 6.60 | 3.4 | 885.9 | 287.4 | 5.30 |
| C WS | 23.0 | 5.58 | 3.7 | 688.5 | 20.6 | 6.08 | 3.5 | 959.0 | 313.6 | 5.70 |
| V WS | 22.4 | 5.54 | 3.6 | 701.9 | 21.1 | 7.04 | 3.4 | 1115.1 | 320.2 | 5.82 |
| *Training system effect* | ns | ns | ns | ns | ns | ns | ns | ns | ns | ns |
| *Water stress effect* | ns | ns | ns | ns | ns | ns | ns | ns | * | * |
| *Significance of training system × water stress interaction* | ns | ns | ns | ns | ns | ns | ns | ns | ns | ns |

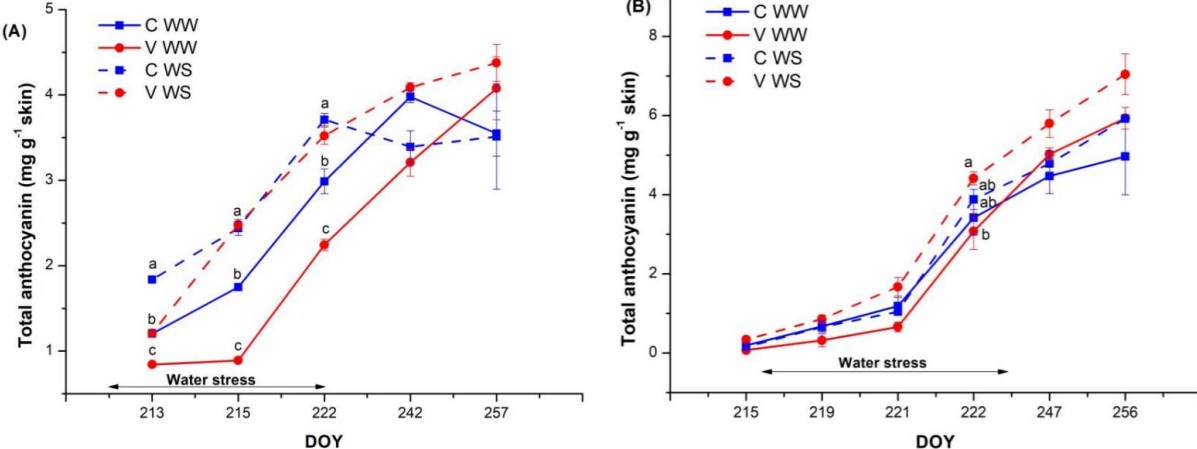

**Figure 4.** Evolution of total anthocyanin concentration (mg g⁻¹ berry skin) in Sangiovese berries sampled in closed (C) and V (V) canopy-shaped vines under normal water supply (WW) or water stress at veraison (WS) in 2018 (**A**) and 2019 (**B**) seasons. Error bars indicate the mean SE (n = 4). Means followed by different letters differ significantly, as calculated by Tukey test ($p \leq 0.05$). The duration of the water stress period is indicated by black double arrows.

In order to understand whether the effect of the canopy and the reduction in water supply at veraison could also affect the concentration of other grape phenols in 2019, we analyzed the concentration of berry flavonols and stilbenes at harvest (Table 7). Regardless of the canopy shape, water-stressed berries showed higher values of flavonols and stilbenes in comparison to well-watered ones. Of note, although not significant, was the detection of

the lowest concentration in C WW of all flavonoid and non-flavonoid compounds, which confirmed an inhibition in the phenylpropanoid biosynthetic pathway.

*3.4. Gene Expression Analyses*

The analysis of gene expression was conducted on the main gene involved in the phenylpropanoid biosynthesis (PAL) and on the key genes of the last step of anthocyanin biosynthesis (DFR, LDOX, UFGT and its transcriptional regulator, MYBA1) (Figure 5). In general, the gene expression was more affected by the canopy shape and by the water stress in 2018 than 2019.The expression of PAL1 was affected in 2018 during the water stress in V berries with an increase in comparison to all the other treatments. Immediately before the end of water stress, an increase in expression was detected in water-stressed berries, regardless of the canopy shape (Figure 5A). This tendency was maintained also in 2019 only at the end of the period of stress, even if no statistically differences in gene expression appeared (Figure 5B). No differences in PAL1 gene expression were detected at recovery in either year (Figure 5A,B).

DFR and LDOX are responsible for the last step of the anthocyanin biosynthetic pathway being expressed immediately before UFGT, the key gene in anthocyanin biosynthesis. The trend of gene expression of DFR and LDOX was quite similar in 2018 (Figure 5C,E), with an increase in expression detected first in DFR, then in LDOX in V WS berries during water stress in comparison to all the other treatments. At recovery, an induction of both genes in water-stressed berries was recorded in comparison to unstressed ones. Only in the case of DFR, and not for LDOX, was an increase in V and C WS berries also detected during water stress, a tendency that was also confirmed in 2019 (Figure 5D,F).

MYBA1 is the transcriptional regulator of UFGT, the gene that presides over the last steps of the anthocyanin biosynthesis in grapevine. During water stress, no statistically significant effect was recorded in MYBA1 gene expression, even if a non-significant increase in expression in V berries was detected regardless of the application of stress in both years and a slight peak of expression of UFGT was simultaneously detected in V WS berries in 2018 at the beginning of the treatment (Figure 5G,H). A similar behavior was observed for C berries at the end of the water stress period, before reaching the maximum peak of anthocyanin concentration detected for this treatment. Very interestingly, at recovery, an upregulation of MYBA1-characterized V berries in comparison to all the other treatments in 2018 and a corresponding peak in UFGT expression was detected (Figure 5G,I). In 2019, no effect was detected in MYBA1 gene expression during the water stress period (Figure 5H), even if in UFGT, an increase was detected in V berries (Figure 5J). At recovery in 2019, no differences were detected in either gene (Figure 5H,J).

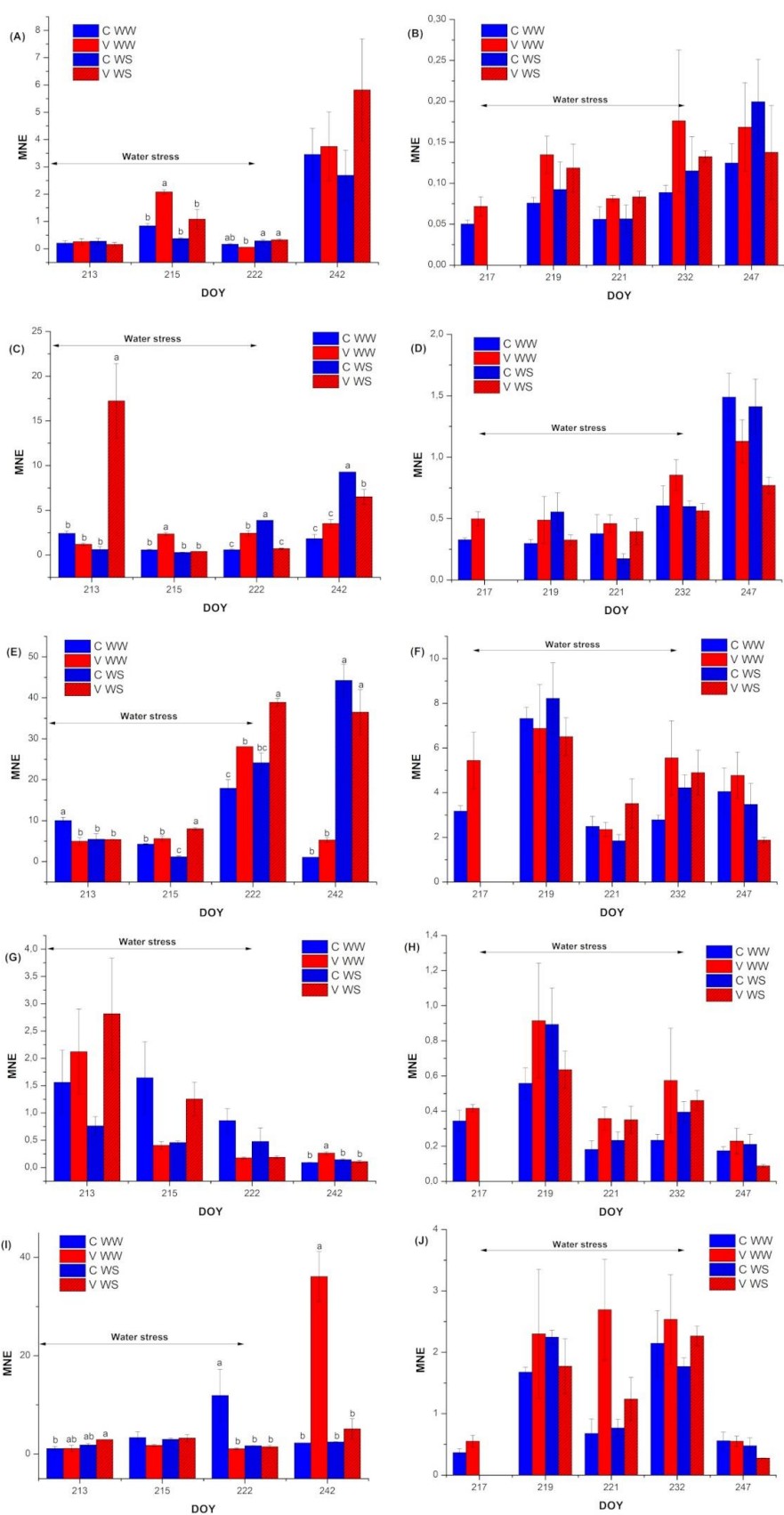

**Figure 5.** Expression profiles of PAL1, DFR, LDOX, MYBA1 and UFGT genes in the skin of Sangiovese berries sampled in closed (C) and V (V) canopy shaped vines under normal water supply (WW) or water stress at veraison (WS) in 2018 (**A**,**C**,**E**,**G**,**I**) and 2019 (**B**,**D**,**F**,**H**,**J**) seasons respectively. Real time

RT-PCR data are reported as mean normalized expression (MNE) values, obtained using actin and ubiquitin-1 as reference genes. Error bars indicate the mean SE (n = 4). Means followed by different letters differ significantly, as calculated by Tukey test ($p \leq 0.05$). The duration of the water stress period is indicated by black double arrows.

## 4. Discussion

### 4.1. Effects of the Shape of the Canopy and of the Water Stress at Veraison on Gas Exchanges, Berry Ripening and Yield at Harvest

Over the two-year trial, the Sangiovese variety was subjected to both a different training system (with closed and open foliage) and a 50% water restriction during veraison. Although the two seasons were characterized by similar climatic conditions, the veraison phase of 2018 was drier than 2019, and the vines underwent greater evaporative demand, which exacerbated the effects of water stress and its duration. Therefore, water limitation during veraison had a greater impact on gas exchange and photosynthetic activity than the shape of the canopy did. As has been established, there are two different mechanisms accredited for the control of stomatal conductance in conditions of water stress, based, respectively, on the biosynthesis of abscisic acid (ABA) or on the variation of leaf turgor pressure [23]. The way in which stomatal conductance is regulated represents the physiological basis of the behavior of the different cultivars in conditions of water stress classified as near-isohydric or anisohydric [22,44]. According to this classification, the Sangiovese variety is considered anisohydric, despite Tombesi et al. [23] having recently shown that the two categories appear not as discrete classes, but as a continuum, and therefore with a different degree of anisohydricity.

Due to the different climatic conditions in the two tested seasons during the water stress period, the vines were subjected to different stress levels: moderate to severe in 2018 and mild in 2019 according to Flexas et al. [45] and Medrano et al. [46], who related photosynthesis to increased levels of water restriction by defining mild, moderate and severe as three distinct stages. The mild phase is characterized by a small reduction in An, and the moderate phase is defined by an increase in the WUEi via stomatal response. If the stress worsens, the gas exchanges are zeroed due to more generalized and dominant non-stomatal limitations to photosynthesis, especially when water stress is accompanied by extremely elevated temperatures and irradiation. Therefore, in 2018, the vines immediately entered a moderate-to-severe stress level, which led to a reduction in both transpiration and photosynthesis and a consequent increase in water use efficiency for WS. In 2018, however the significant reduction in stomatal conductance halved the assimilation, whereas in 2019, conductance values well above 100 mmol m$^{-2}$ s$^{-1}$ caused only a slight decrease in photosynthesis for the WS compared to WW. Although the water stress caused a significant decrease in gas exchange, this result was not found for the two forms being compared. Several researchers have asserted that in non-water-limiting conditions, the amount of light intercepted by the vine canopy is important in determining the overall water consumption [47,48]. Water consumption was higher when the canopy could grow unrestricted (open hedge) than when the canopy was limited (compact hedge), since the latent heat flux intercepted by the open canopy was greater than the compact hedge [49]. Other authors such as Valentini et al. [27] have instead reported that in conditions of both thermal and radiative stresses, the vines grown with closed and open foliage displayed a different behavior in relation to water loss.I It has been reported in detail that in the hottest hours of the day, the open-canopy vines reveal both a photosynthetic depression and a decrease in transpiration activity. Further, non-stomatal limitations such as photoinhibition and feedback inhibition through source–sink interactions have been suggested as important limiting factors in a multiple summer stress scenarios [50]. Conversely, in our conditions, the vines were not subjected to thermal stresses such as zeroed gas-exchange activities.

As has been established, veraison is widely considered a crucial stage for solid soluble accumulation in berries, and many factors can affect their rate of growth, such as temperature [51,52], irradiance [53,54], water content [55,56] and canopy management [27,39,57].

As reported by Castellarin et al. [58], water restriction on Cabernet Sauvignon accelerates ripening and affects berry mass. In effect, the water deficit before veraison increased the initial rate of sugar accumulation and reduced berry weight [58].

In our case, we did not find any difference in either soluble solids or berry weight, perhaps because the reduction in irrigation was limited to a few days during veraison when berry size was less sensitive to water stress [59]. These results are consistent with those obtained by partial root drying (PRD) on potted Tempranillo [60] and with those obtained in unirrigated Syrah [61] (Schultz, 1996).

Raynolds and Heuvel [57] stated that the training system is the physical manipulation of a plant's form. Several studies have also highlighted how the modification of the total amount or the distribution of the leaf area through the training system alters both the microclimate of the vine and its performance [27,31,62]. Though Howell [63] reported that a range of 7 to 14 cm$^2$ of total leaf area per gram of fruit is required to reach fruit maturity, Dookozlian and Kliewer focused on the ratio of exposed to unexposed leaves, a relationship that is directly influenced by the training system [53]. Starting from these assumptions, all the variables of yield and fruit composition are strongly dependent on the photosynthetic activity of the leaves. Therefore, in our conditions, the absence of differences between the canopy shape in terms of yield attributes and soluble solids at harvest is strictly related to the absence of differences in photosynthesis.

### 4.2. Effects of the Canopy Shape and of the Water Stress at Veraison on Anthocyanin Accumulation during Ripening and on Polyphenol Compounds at Harvest

Environmental conditions have a strong impact on anthocyanin concentration and composition, and their accumulation appears to be more susceptible to adverse climatic conditions than that of other compounds that characterize berry composition [54]. Guidoni et al. [64] demonstrated that anthocyanins accumulate in a two-phase process, with an initial stage influenced mainly by vine vegetative conditions and cultural practices and a second phase strongly affected by climate. During anthocyanin accumulation, heat and radiation have a synergistic effect at moderate temperatures and an antagonistic effect at high temperatures [13,16], and water deficit could result in an increase in anthocyanin concentration depending on the timing and duration [65–67]. In much research, the increase in anthocyanin concentration in water-stressed berries has been linked to a decrease in berry weight, but no indications are present on the effect of the canopy shape. In our conditions, since neither water stress at veraison nor canopy shape affected berry weight, we could assume that the changes detected in the anthocyanin accumulation were linked to different modulations of synthesis and degradation processes due to the different conditions applied. As many other researchers have reported, veraison is the key phase for the triggering of anthocyanin accumulation. This aspect is further confirmed by the results of the present research, in which, despite the differences in the climatic conditions of the seasons, a stable trend of anthocyanin accumulation was detected.

During veraison, water stress had a major effect on anthocyanins when compared to the canopy shape. This could be related to an increase in the expression of key genes involved in anthocyanin biosynthesis on the direct action of the water stress in association with the role of hormonal signals, in particular of ABA, a drought-inducible hormone that stimulates the anthocyanin biosynthesis [68,69].

However, in the absence of water stress at veraison, the canopy shape resulted in a differentiation of the accumulation of anthocyanins, probably due to the microclimatic conditions to which the clusters were subjected. The berries in the V open canopy were directly exposed to sunlight during the warmest hours of the day (around midday), which could have delayed the initial accumulation of the anthocyanins, as temperature can be also prohibitive for anthocyanin accumulation. The clusters within a closed canopy were instead characterized by a different exposure to sunlight. In a north-south-oriented row, as seen in our conditions, there were clusters exposed to the east in the early morning and to the west in late afternoon. For these reasons, the microclimate conditions of the clusters

present in a well-watered vine with a closed canopy seemed to be in better condition for anthocyanin accumulation compared to the ones in the V open canopy.

After veraison, the reestablishment of water conditions partly modified the previous statements. In addition, no differences were found between the treatments for 2018 at harvest, though an unexpected trend was observed for an increase in anthocyanins in the berries of the V-shaped canopy compared to C regardless of the previous imposition of water stress. In order to explain this behavior, we analyzed the seasonal conditions of both years post-veraison. In 2018 and 2019, we registered maximum temperatures below 30 °C after verasion, which are optimal for anthocyanin biosynthesis in Sangiovese [15]; however, light exposure is required in addition to optimal temperature to promote anthocyanin synthesis [70]. Our results suggest that the meteorological conditions to which the berries were subjected in the open canopy could have favored anthocyanin accumulation after veraison due to optimal temperatures and more uniform exposure of the bunches in comparison to those that characterized the berries within the closed canopy. Furthermore, the typical trend in total anthocyanin accumulation in the grapevine usually involves a rapid increase until they became stable or even decline [71], and the early achievement of a peak in anthocyanin accumulation in the berries of the closed canopy prevented further accumulation.

In the literature, it has been reported that stilbene accumulation is mainly affected by the cultivar [72–74] and, to a lesser extent, by water deficit. In fact, Savoi et al. [75] reported that a water deficit may also decrease the biosynthesis of these compounds, as competition occurs between the stilbenoid and flavonoid pathway.

Some studies have shown that water deficit had no effect on flavonol concentration [75,76], and others have reported an increase in these compounds during berry ripening [77–79]. In our experimental conditions, water stress had a major effect on the increase in concentration of both flavonols and stilbenes.

### 4.3. Effects of the Canopy Shape and of the Water Stress at Veraison on the Expression of the Genes Involved in Anthocyanin Biosynthesis

The effect of water stress and of the microclimate surrounding the clusters on anthocyanin biosynthesis genes has been extensively studied, and many researchers agree that the increase in anthocyanin concentration following water stress is driven by an increase in the expression of the corresponding genes [75,80]. However, the different canopy shape and the cluster's microclimate effects on the modulation of genes involved in anthocyanin biosynthesis has not been widely studied. In previous works, we reported that the exposition of clusters to sunlight or to maximum temperatures higher than 35 °C had a deleterious effect on the expression of the genes involved in anthocyanin biosynthesis and in the accumulation of such compounds in the Sangiovese variety [15,16,39].

In our conditions, we can hypothesize that water regimes and canopy shape act differentially on anthocyanin biosynthesis gene expression. At veraison, severe water stress had a major effect on the expression of the genes, triggering higher anthocyanin biosynthesis due to differences in the expression of the anthocyanin biosynthetic genes detected in 2018, when all the genes except MYBA1 were differently modulated among treatments during the water stress period. In particular, the water stress treatment was able to induce the expression of PAL1, DFR, LDOX and UFGT in at least one sampling date, and for some genes this effect had consequences even afterward, when the water conditions became similar again (as in the case of DFR and LDOX). Sangiovese is traditionally classified as a near-anisohydric cultivar and the stomatal sensitivity to ABA has a less pronounced result than that of grapevine cultivars that instead exhibit a near-isohydric behavior [81]. For this reason, we can hypothesize that in Sangiovese, the ABA-dependent pathways that have been demonstrated to possibly regulate the phenolic biosynthesis during ripening even under non-stressed conditions [82,83] could coexist with ABA-independent pathways activated specifically in response to water deficit, thus resulting in the detected gene expression increase at veraison. Though the microclimate conditions of the bunches in

the V canopy might have been less favorable to anthocyanin accumulation, at recovery, MYBA1 and UFGT showed an increase in the berries of the well-watered V canopy, which allowed it to reach the same anthocyanin concentration detected in the other treatments as previously described.

The different climatic conditions of 2019 that allowed for a mild water stress caused an overall delay in the appearance of gene expression differences and, when present, their quantity was much lower than those detected in 2018. For this reason, it was difficult to detect stable differences in gene expression, which confirms what we had previously discussed about the dominant effect of the water stress at veraison compared with the canopy shape.

## 5. Conclusions

In the wine-growing areas characterized by average temperatures around 30 °C in summer and no rainfall, the presence of water stress is increasingly frequent. When Sangiovese, a variety considered near-anisohydric, is subjected to a moderate-to-severe stress during veraison, a strong effect on the physiology of the vine could occur, leading to a decrease in the transpiration rate and a reduction in photosynthetic activity. However, this phenomenon is less evident when the water stress at veraison is mild, and it is absolutely unrelated to the shape of the canopy in both years. Neither the water stress limited to veraison nor the canopy shape are able to influence the yield parameters and the accumulation of soluble solids. Although an induction of the expression of genes involved in anthocyanin biosynthesis in moderate-to-severe stress leads to an increase in anthocyanin concentration during veraison, no solid effects were detected after re-watering or at harvest. In conclusion, the effects of water stress during veraison on both the physiology and berry ripening are subject to environmental and microclimatic conditions, and it is hypothesized that the shape of the canopy can play a role under elevated temperature and/or prolonged water restriction.

**Author Contributions:** Conceptualization, G.V., C.P. and I.F.; Data curation, G.V., C.P. and G.A.; Formal analysis, G.V., C.P., G.A., R.M. and F.C.; Supervision, I.F.; Writing—original draft, G.V., C.P. and R.M.; Writing—review and editing, G.A., F.C. and I.F. All authors have read and agreed to the published version of the manuscript.

**Funding:** This research received no external funding; it was part of the Ph.D program of the University of Bologna.

**Institutional Review Board Statement:** Not applicable.

**Informed Consent Statement:** Not applicable.

**Data Availability Statement:** Not applicable.

**Acknowledgments:** The authors thank Eugenio Magnanini for his help in setting up the lysimeters.

**Conflicts of Interest:** The authors declare no conflict of interest.

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
