# Peer review of "Vine Physiology, Yield Parameters and Berry Composition of Sangiovese Grape under Two Different Canopy Shapes and Irrigation Regimes"

_agronomy, doi:10.3390/agronomy12081967_

Round 1

Reviewer 1 Report

The submitted manuscript is clearly and concisely written. Relevant literature is listed. I think that the experimental part, which refers to the analysis of anthocyanins and flavonols, should be supplemented. Namely, the authors cite reference 8 (Mattivi, F.; Guzzon, R.; Vrhovsek, U.; Stefanini, M.; Velasco, R.) as a method for anthocyanin extraction. These authors cite other references for it. Therefore, it seems to me better to give a short procedure with reference. I also do not see a method for determining total anthocyanins, flavonols and stilbenesThe submitted manuscript is clearly and concisely written. Relevant literature is listed. I think that the experimental part, which refers to the analysis of anthocyanins and flavonols, should be supplemented. Namely, the authors cite reference 8 (Mattivi, F.; Guzzon, R.; Vrhovsek, U.; Stefanini, M.; Velasco, R.) as a method for anthocyanin extraction. These authors cite other references for it. Therefore, it seems to me better to give a short procedure with reference. I also do not see a method for determining total anthocyanins, flavonols and stilbenes

Author Response

Dear Editor,

we would like to thank you and the reviewers for the suggestions to improve our paper entitled “Vine physiology, yield parameters and berry composition of Sangiovese grape under two different canopy shapes and irrigation regimes”.

We agree with all their comments and we have revised our paper accordingly.

We included a point-by-point response to the issues raised by the referees, the revised version of the manuscript with the changes easily evident and the modified figures on editorial submission system.

We are confident that the last version of the manuscript has been improved, and may now satisfy the requirements for publication in Agronomy.

If you need any further information, please do not hesitate to contact me.

Yours sincerely,

Chiara Pastore

We report here the Reviewer comments and our point by point response in bold:

Reviewer 1

The submitted manuscript is clearly and concisely written. Relevant literature is listed. I think that the experimental part, which refers to the analysis of anthocyanins and flavonols, should be supplemented. Namely, the authors cite reference 8 (Mattivi, F.; Guzzon, R.; Vrhovsek, U.; Stefanini, M.; Velasco, R.) as a method for anthocyanin extraction. These authors cite other references for it. Therefore, it seems to me better to give a short procedure with reference. I also do not see a method for determining total anthocyanins, flavonols and stilbenesThe

We thank the reviewer 1 for the suggestion and we integrate the Material and methods accordingly (lines 129-134)

Reviewer 2 Report

Review on the manuscript titled ‘Vine physiology, yield parameters and berry composition of Sangiovese grape under two different canopy shapes and irrigation regimes’

In this manuscript, the authors attempt to characterize the effect of canopy shapes and water stress on grape physiology and berry ripening. Overall, the work is well done (experimental plan and procedures) and the manuscript is well written. I have only some few concerns that are detailed below.

1. The position of Figure 1 is not aligned with Table 1, furthermore, Figure 1 is not aligned with its legend. Please adjust the layout.

2. The legend of Figure 2 has a ‘.’ in the end, while other figure legends have ‘..’ in the end.

3. ‘DOY’ is a commonly used word throughout this study, but I have not seen its full English name, which is not conducive for readers to understand its meaning. So it is suggested that the full name of this word should be used when it first appears in the paper, and then the abbreviation can be used in the bracket.

4. The legend of Figure 2 demonstrates that ‘The duration of the water stress period is indicated by black double arrows’, however, I could not find the sign of the double arrows in Figure 2, although I did find a black double arrow in Figure 1.

5. In Table 3 to Table 6, since WS (water-stress) and WW (well-watered) are two corresponding conditions, when dealing with group naming, the authors should not only name C/V WS, but also name C/V as C/V WW.

Author Response

Dear Editor,

we would like to thank you and the referees for the suggestions to improve our paper entitled “Vine physiology, yield parameters and berry composition of Sangiovese grape under two different canopy shapes and irrigation regimes”.

We agree with all their comments and we have revised our paper accordingly.

We included a point-by-point response to the issues raised by the referees, the revised version of the manuscript with the changes easily evident and the modified figures on editorial submission system.

We are confident that the last version of the manuscript has been improved, and may now satisfy the requirements for publication in Agronomy.

If you need any further information, please do not hesitate to contact me.

Yours sincerely,

Chiara Pastore

We report here the Reviewer comments and our point by point response in bold:

Reviewer 2

In this manuscript, the authors attempt to characterize the effect of canopy shapes and water stress on grape physiology and berry ripening. Overall, the work is well done (experimental plan and procedures) and the manuscript is well written. I have only some few concerns that are detailed below.

  1. The position of Figure 1 is not aligned with Table 1, furthermore, Figure 1 is not aligned with its legend. Please adjust the layout.

We thank the reviewer 2 for the suggestion, the layout has been adjusted

  1. The legend of Figure 2 has a ‘.’ in the end, while other figure legends have ‘..’ in the end.

We have standardised the legend in all the figures

  1. ‘DOY’ is a commonly used word throughout this study, but I have not seen its full English name, which is not conducive for readers to understand its meaning. So it is suggested that the full name of this word should be used when it first appears in the paper, and then the abbreviation can be used in the bracket.

We have included the full name with the abbreviation in the bracket (line 96)

  1. The legend of Figure 2 demonstrates that ‘The duration of the water stress period is indicated by black double arrows’, however, I could not find the sign of the double arrows in Figure 2, although I did find a black double arrow in Figure 1.

We thank the reviewer and we have removed the sentence: The duration of the water stress period is indicated by black double arrows’ including the following sentence “during the period of stress” (line 263)  

  1. In Table 3 to Table 6, since WS (water-stress) and WW (well-watered) are two corresponding conditions, when dealing with group naming, the authors should not only name C/V WS, but also name C/V as C/V WW.

We thank the reviewer for the useful suggestion and we modify accordingly the Tables and the Figures legends

Round 2

Reviewer 2 Report

The authors have made detailed improvements based on the previous comments, and the current version of the manuscript possess the publishable quality. Therefore, I agree to accept and publish this study.

Author Response

We thank again the reviewer for the useful comments and suggestions.